# Turning Crisis into Opportunities: How a Firm Can Enrich Its Business Operations Using Artificial Intelligence and Big Data during COVID-19

**Yasheng Chen [1] and Mohammad Islam Biswas [2,3,*]**

1   Department of Accounting, School of Management, Xiamen University, Xiamen 361005, China; yshchen@xmu.edu.cn
2   Institute of Financial and Accounting Studies, School of Management, Xiamen University, Xiamen 361005, China
3   Department of Accounting, School of Business, Bangladesh University of Business and Technology (BUBT), Dhaka 1216, Bangladesh
*   Correspondence: biswas8042@gmail.com

**Abstract:** The COVID-19 pandemic has severe impacts on global health and social and economic safety. The present study discusses strategies for turning the COVID-19 crisis into opportunities to use artificial intelligence (AI) and big data in business operations. Based on the shared experience and theoretical ground, researchers identified five major business challenges during the COVID-19 pandemic: production and supply-chain disruption, appropriate business model selection, inventory management, budget planning, and workforce management. These five challenges were outlined with eight business cases as examples of companies that had already utilized AI and big data for their business operations during the COVID-19 pandemic. The outcomes of this study provide valuable insights into contemporary social science research and business management with AI and big data applications as a business response to any crisis in the future.

**Keywords:** artificial intelligence; big data; business operations; COVID-19 pandemic

## 1. Introduction

The COVID-19 pandemic hits public health as well as the world economy, and people around the world now prepare to lead their lives in a challenging environment as a new normal era of the world [1]. Since the influenza of 1918, the COVID-19 pandemic has been the worst hazard to social life, public health, and the global economy [2]. The COVID-19 crisis differs significantly from past crises regarding its severity, cause, and scope [3]. Governments worldwide have taken unprecedented measures such as maintaining lockdowns, advocating social distancing, urging people to wear face masks, and encouraging hygiene practices to minimize the spread of this infectious virus.

Although public health is the worst-hit sector, the world economy is now most distressed by COVID-19 [2]. This crisis has severe adverse impacts on all business sectors, such as tourism, hospitality, transportation, manufacturing, and trading companies. Along with the adverse consequences of COVID-19 on public health [1,2,4], the pandemic has also disrupted the smooth business operations and has faced businesses with some challenges [5–7]. Therefore, the query arises what challenges business face during the pandemic and how to deal with these challenges to adjust to the new normal age? So far, to the best of our knowledge, up to date, no study has focused on the business challenges and their possible solutions using AI and big data in business operations during this pandemic. Thus, the present study finds a research gap stating accepted evidence on how a firm can turn the COVID-19 crisis into an opportunity using AI and big data.

Recalling the proverb "change always creates new and different opportunities" [8], the present study replies to the above proverb by addressing two disruptive technologies

of Industry 4.0, such as AI and big data, in the contemporary business plan for smooth business operations during the COVID-19 pandemic. AI facilitates the automation of various business processes across different industries and can break through the limited boundaries of human data collection and data processing capabilities for decision making [9]. In the era of technological advancement, data is referred to as the new oil [10], while AI is frequently referred to as the new electricity capable of extracting value from this oil. For example, food suppliers were facing abrupt order cancellations from their customer bases at the beginning of COVID-19. It is widely understood that COVID-19 transfers from person to person. Thus, customers want the daily necessities provided by contactless delivery. Introducing advanced predictive analytic tools, such as AI and big data, improved hygiene and required fewer face-to-face interactions between workers and the firm's representative and customer. The above example is typically a challenge and possible solution to supply chain disruption during the pandemic.

In this way, with the shared experience of different business cases, researchers attempt to identify the major challenges of business operations during the pandemic. Specifically, this study first identifies five important challenges for effective business operations during COVID-19: production and supply chain disruption, appropriate business model selection, inventory management, budget planning, and workforce management. These five challenges were outlined with eight business cases as examples of companies that had already utilized AI and big data for their business operations during the current pandemic. On the other hand, in light of the resource-based view theory (RBVT), AI and big data are recognized as invaluable assets and time-driven strategic tools of a contemporary business plan for a possible solution to mitigate the identified challenges for business operations during the pandemic. That is why this study is an insightful view of AI and big data as critical issues that deserve a systemic discussion.

## 2. Theoretical Background

Originally, a resource-based theory, offered by [11], with a hypothesis that economic entities have various resources (tangible and intangible), and firms' overall performances (financial and social) are directly contingent upon a firm's capacity to utilize of these invaluable resources in the changing environment. Later, Penrose's propositions were stretched and adapted into an insightful view of the resource-based theory, now popularly known as the resource-based view theory (RBVT), by labeling business resources into two groups such as heterogeneous and immobile assets [12]. Several scholars recommend that the firms' capacity to fully utilize these diversified resources is a competitive advantage and risk mitigation weapon in any challenging environment [13,14]. Thus, based on Barney's RBVT, the present study proposes AI and big data as invaluable business resources and strategic tools to mitigate business operation challenges during the pandemic.

### 2.1. Artificial Intelligence (AI)

Over the past few decades, the topic of AI has attracted impressive attention from business leaders. The Dartmouth Research Project defined AI as the problem of "making a machine behave in ways that would be termed intelligent if a human being behaved like this." AI can function intelligently and, in ever greater regions to do so [13], perceive external data accurately, think, and learn in the same way as a human being [15]. AI is gradually conquering our reality and predicting the future of how firms will be organized and controlled [9], ranging from inventory management, business model selection, workforce management, and budgeting to supply chain management. Therefore, academic scholars and business leaders have reached a consensus that AI would significantly change the way of smooth business operation. For example, the Xerox Services technology company applied an AI recruitment algorithm to aid hiring managers, giving them a snapshot of how well the skills of candidates are suitable for jobs.

AI is looked at through the lens of a firm's capability rather than technology [9]. From a broad viewpoint, AI can care about several vital needs in the business operations, such

as data analysis in business activities, automating business processes, and engaging with primary stakeholders such as employees, customers, and suppliers [16]. For example, when compared with conventional budgeting, AI and machine learning-enabled budgeting is automatically generated, monitored, controlled, and analyzed by AI algorithms, the budget variance can be detected faster and thus readjusted in a timely manner. Several business legends have started recognizing AI's and machine learning's potential and have begun incorporating them into their operating process. As a strategic management tool, AI smooths the development of new models, forms, means, system architectures, and technology systems in the domain of intelligent business operation in the challenging business environment [17].

### 2.2. Big Data

In the era of technological advancement, data are considered one of the most valuable issues in the automation process of businesses [10]. In recent decades, business and technological growth have been connected with human activity. Therefore, big data has become part of everyday business and continues to stimulate potential creativity. Currently, most business fields are linked to big data, including the manufacturing process, supply chain management, sales, marketing, B2B process, and R&D management [18,19].

Big data is one of the latest business and technological ideas in the new generic era of the world with the presence of the COVID-19 [13]. The applications of big data work as measures of the ability of a company's innovation ability to respond to business opportunities. For instance, big data are using analytic capabilities to educate business responses in facing the challenges of COVID-19 and planning for the future [20]. These data types are reachable universally but cannot be succeeded straightforwardly due to their complexity. It has grown in eminence amongst firms in the recent few decades due to its paybacks and the potential of shaping strategic values through business operations [14]. The contemporary research lineup highlights the positive effects of the capacity to use big data in their respective organizational business processes [13]. Therefore, the ability to use big data can quickly transform and adapt business processes to challenges and opportunities.

### 2.3. Conceptual Framework

The economic shocks of companies due to COVID-19 have resulted in historic moments of truth. However, companies can change the business background and the conditions of competition quickly, often in an unobtrusive way that was not apparent earlier [2]. Companies that take bold steps in times of difficulty may turn adversity into advantages [13]. For example, the 2003 SARS pandemic is often credited with establishing e-commerce giants such as JD.com and Alibaba in China, while Starbucks and American Express turned to digital operation during the 2008–2009 financial crisis that dramatically raised shareholder's value. The COVID-19 pandemic will accelerate several changes that have already begun, and companies are moving towards regaining their focus. For instance, instead of focusing heavily on sourcing and producing some low-cost locations, businesses are creating more flexibility in their price chain. Therefore, consumers are buying more goods and services online, and many people are working remotely.

Historically, automation happens when economic difficulties are combined with maturing technology [21]. In the economic difficulties caused by COVID-19, AI and big data can help companies adopt the digitalization trend. Through advanced analytic and machine learning, AI and big data can enable companies to identify new customers and deliver goods and services to online buyers. Advanced robots capable of recognizing and handling items previously handled by a human would enable the 24-h operations of plants. AI-based technologies can help businesses stimulate a live working environment better and build a workforce on demand [22]. Thus, AI technologies can bring development and changes into the business industries.

The COVID-19 pandemic has created new challenges (e.g., production and supply chain disruption, appropriate business model selection, inventory management, budget

planning, and workforce management) for the smooth operations of a firm. However, the two disruptive technologies of Industry 4.0, such as AI and big data, show the way of smooth business operations during the COVID-19 pandemic. AI extends the cognitive utility and enhances human working capabilities, including natural language processing, machine learning algorithms, and computer vision. Big data has high velocity, high volume, and a comprehensive variety of information resources [23]. Thus, business leaders can take a practical decision with the help of these two disruptive Industry 4.0 technologies for continuous business operations during any crisis such as COVID-19. Figure 1 shows the new reality/challenges and how AI and big data can support a firm during COVID-19 (details discussed in part 4).

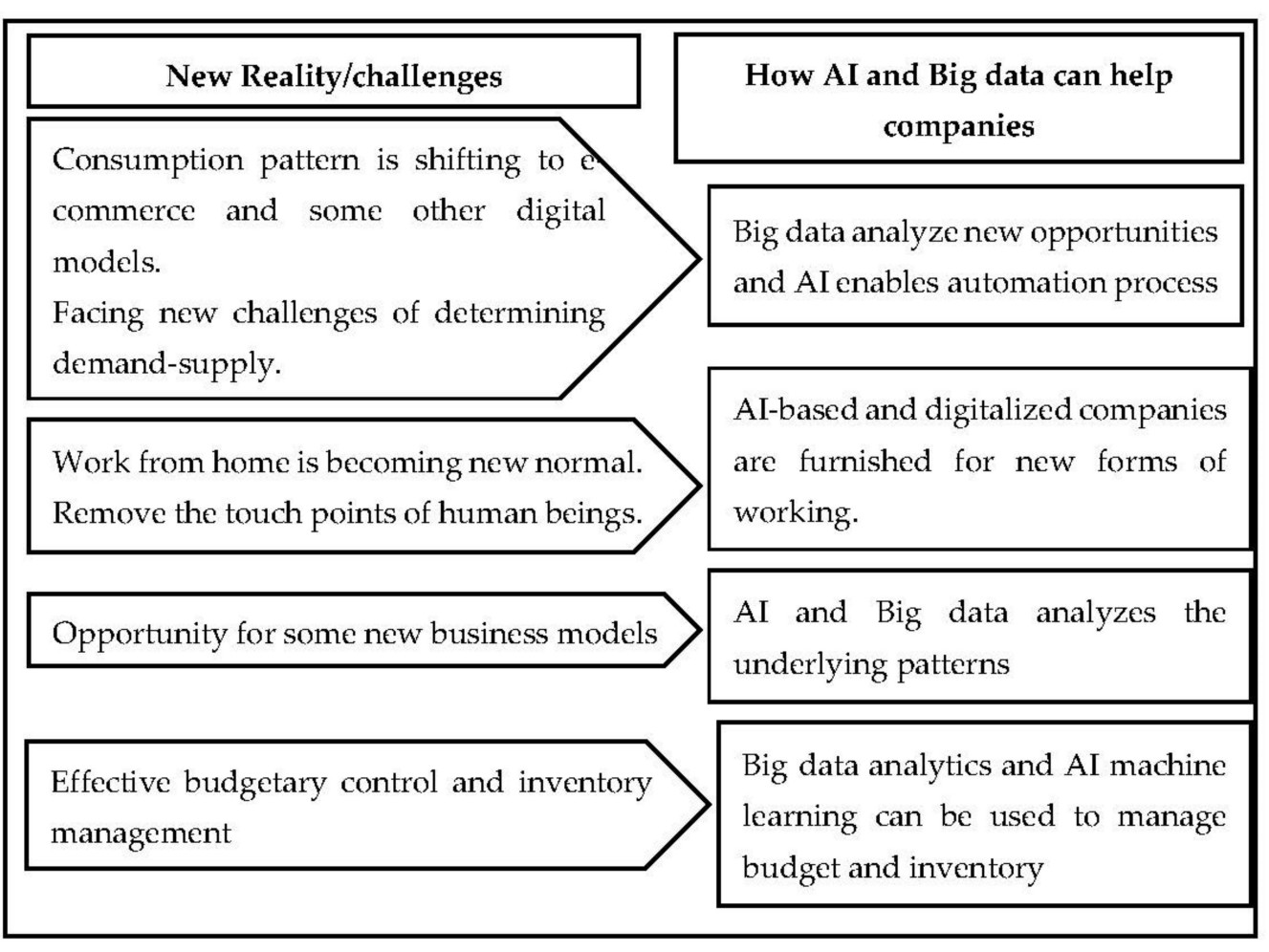

**Figure 1.** Usages of AI and Big data for continuous business processes in the changing environment.

### 3. Research Methodology

#### 3.1. Selection of Research Area

This study is intended to review manual content analysis following [2,24] that focuses on the immediate response of using AI and Big data in the business operations to the current pandemic. As we already know, in late December 2019, in the human body in Wuhan, China, an unknown infectious disease was detected, which resulted in the lockdown of most of China's cities to stop the infectious virus's transmission. However, during lockdown across countries, people limited their mobility, thereby causing great economic pain and extreme severity in various industries with travel bans affecting the supply-chain disruptions, cancelation of new orders, affecting the manufacturing industry, and a ban on mass gathering creating scarcity in the workforce [7]. For example, the

contraction of the Chinese economy in 2020 experienced a loss in labor output of around 4%, global output declines of 1%, and a total monthly economic loss of USD 27.309 billion during the lockdown [25]. In that context, the adaptation of AI and big data were likely to have a significant social and economic impact across countries since these Industry 4.0 technologies were the only path forward for regular business operations with minimal human intervention. In addition, consumer behavior in the lockdown has influenced improved growth and development of AI technologies. Based on these perspectives, the research area and the use of AI and big data for smooth business operation during the pandemic are specified and considered for further findings.

### 3.2. Sample Selection

Researchers attempted to gather data to meet the objective of this study through Internet access. First, through visiting numerous companies' webpages and related news links, researchers captured the shared experiences and identified the major challenges for firms' business operations during the pandemic. Second, researchers followed firms that use AI and big data to mitigate the identified challenges for the respective firms' business operations. In the light of the judgmental sampling procedure, the eight companies were selected to represent AI and big data users during the COVID-19 pandemic. The sample firms of this study are proposed in Table 1.

**Table 1.** Selected firms who used AI/big data during the COVID-19 pandemic.

| S.L | Company and Its Type | Impact on Revenue/Service of Using AI/Big Data | Sources |
|---|---|---|---|
| 1 | Meituan-Dianping, a Food delivery company | 80% delivery completed via zero-contact model | https://www.bcg.com/publications/2020/how-chinese-digital-ecosystems-battled-covid-19 (accessed on 22 April 2021) |
| 2 | Forest Cabin, Cosmetics company | Revenue rose 120% over the same period of last year | https://www.bcg.com/publications/2020/how-chinese-digital-ecosystems-battled-covid-19 (accessed on 22 April 2021) |
| 3 | Lin Qingxuan, a Chinese cosmetic company | Revenue increased by 200% in Wuhan compared to the previous year | www.thepourquoipas.com/post/covid-business-model-innovation-in-china (accessed on 26 April 2021) |
| 4 | JD.com, e-commerce | 0.2% sales increased in China during April compared to the previous year | [21] |
| 5 | Toyota Motor, Car manufacturer | Received and delivered 10 billion new orders | https://english.kyodonews.net/news/2020/06/eb387d16f9a3-new-car-sales-of-toyota-surge-in-china-in-may-amid-easing-virus-fears.html?phrase=japan&words= (accessed on 2 May 2021) |
| 6 | Master Kong, noodles producer, and distributor | Received and delivered 10 billion new orders | www.bcg.com/publications/2020/how-chinese-digital-ecosystems-battled-covid-19 (accessed on 22 May 2021) |
| 7 | Ideal, Jewelry company | Sales rose 40% during the pandemic | www.thepourquoipas.com/post/covid-business-model-innovation-in-china (accessed on 27 May 2021) |
| 8 | Mater Hospital, Ireland, Hongshan Sports Center, a field hospital in Wuhan, China | Saved three hours of work every day Treated over 20,000 patients | https://www.uipath.com/newsroom/uipath-launches-a-pro-bono-automation-project-with-the-mater-hospital (accessed on 3 May 2021) https://www.cnbc.com/2020/03/18/how-china-is-using-robots-and-telemedicine-to-combat-the-coronavirus.html (accessed on 3 May 2021) |

### 3.3. Analytical Approach

The present study is grounded on a qualitative research method in the research lineup of [2,13] as data retrieved from companies' webpages and related news links [2,26]. This research method has already been used to inspect the application of technologies during the COVID-19 pandemic [27], CSR practices to HRM [26], the communication strategies of

Fortune 100 companies [28], and the role of newspapers in influencing social policy after Hurricane Katrina [29].

In this study, the researchers read each document carefully to identify the actions taken for continuing their business operations smoothly. Next, the data were categorized into firms that used AI and big data during the pandemic. In this way, the objective of the study is accomplished by identifying five major challenges of firms' business operations during the pandemic and proposing AI and big data (in light of eight cases that have already utilized AI and big data) as strategic tools to turn the COVID-19 crisis into an opportunity for business operations.

*3.4. Research Framework*

The content analysis approach of this study contains data from secondary sources (see Table 1 for sources) following the research of [27]. A systematic process is followed from data searching platforms to research area selection and business challenges selection to an expected solution of this study. The overall methodological framework is shown in Figure 2. The idea of this framework has been followed by [2,13,24].

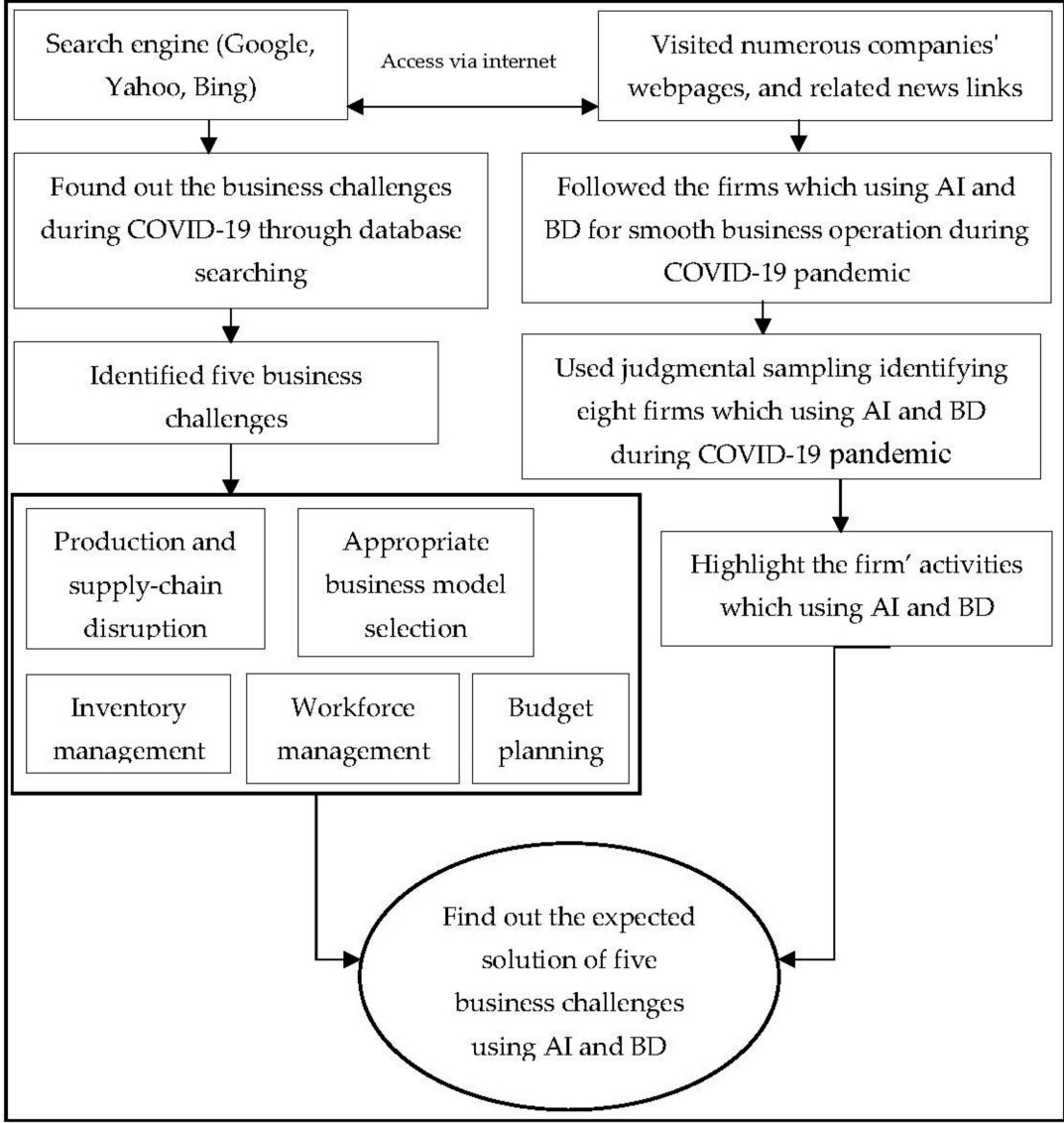

**Figure 2.** Research framework (Source. Authors' illustration), Note: AI = Artificial Intelligence, BD = Big data.

## 4. Major Challenges and Possible Strategic Solution

This section discusses the impacts of five major business challenges that firms faced during the COVID-19 crisis: production and supply chain disruption, appropriate business model selection, inventory management, budget planning, and workforce management. It also describes the use of AI and big data as possible solutions by explaining eight business cases of respective firms that have already utilized AI and big data as strategic I4.0 technologies to mitigate the respective challenges during the COVID-19 pandemic.

*4.1. Major Challenges for the Business Operations during the COVID-19 Pandemic*

4.1.1. Challenges for Production and Supply-Chain Management

The COVID-19 pandemic is not the first natural disaster that has suddenly disrupted the global supply chain. Other natural disasters, including the 2011 earthquake in Japan, the 2004 Indonesian tsunami, and the SARS outbreak in China in 2003, led to a shortage of necessary products. However, the impacts of this virus are different in terms of their scope and severity. Over the last few months, this infectious virus has sped around the globe, sending millions of people into lockdown and into maintaining physical distancing [30]. In this way, the pandemic disrupts not only the local supply chain but also the global supply chain at all levels, from supply sources to final users [31]. The production process is much more complicated compared to a few decades ago. Different components are required to be sourced from other parts of the world, and then finished goods need to be transported worldwide. The over-dependence on logistics creates a problematic position for imports, exports, and production when supply chains are disrupted. When in January 2020, China sent her workers home and stopped businesses to minimize the pandemic, the steady flow of essential parts to the global supply chain stopped or slowed. For example, the COVID-19 pandemic is having a pronounced impact on the supply chains of almost all wholesalers, retailers, and manufacturers as well [7].

Researchers have also conducted various simulations and optimization supply chain models in order to resolve uncertainties, for example, a digital supply chain framework for risk management during the pandemic [32]. However, production and supply chain activities are still not discussed to apply AI and big data for smooth business operation during any crisis. Firms face new challenges in mismatched demand and supply and in improving a stable supply chain during the pandemic [7]. Manufacturers in key industries scrambled to search for alternative suppliers to keep their factories running. When some regions crossed the most severe stage of the pandemic and began a recovery plan, management teams examined the supply chain's reliability and risk. Products are distributed through some exclusive challenges, such as some online channels. The lockdown and curfew situations across the globe have led to a peculiar scenario where the demand is only for essential items, and luxury items are facing significant challenges.

4.1.2. Challenges for the Business Model and Innovation

In addition to the impact on public health, COVID-19 has significantly impacted consumer behavior, attitudes, and purchasing habits [33–35]. Nobody was ready for what the pandemic brought to the world. When COVID-19 started to spread, managers began to think of how it would impact their business models [6], such as in the supply chain, managing inventory, employee well-being, and business continuity. No quick solution was available, and organizational value chains were threatened. The current crisis indicates that traditional business models are no longer useful or cannot guarantee smooth business operations as they used to before the COVID-19 pandemic. Social distancing has driven some firms out of business (e.g., tourism, airlines) and has forced others to reorganize their operations to remain afloat. Similarly, firms have to shift their business models due to social isolation, launch new goods/services, reposition their existing activities to fit with the lockdown lifestyle, and adopt contactless distribution methods [36]. However, a crisis often addresses new business ideas that include new value propositions, capabilities, and customer needs.

The COVID-19 and its containment strategies have changed the way we work and consume. Consumers' purchasing behavior and demand have fundamentally changed during the crisis, making it more critical for businesses to rely on a new innovative marketing strategy to survive [37,38]. Consumers are dramatically cutting their most discretionary spending, severely impacting some industries such as entertainment, restaurants, travel, apparel, accessories, and footwear. With the emergence of new consumer behavior, companies have unique opportunities to expand their existing offerings and accelerate the digital business by creating new business lines. These developments may force companies to review their new business models and innovate to capture new marketplaces and digital customer segments.

### 4.1.3. Challenges for Inventory Management

All economic sectors are linked through a complex supply chain network, but activities have been put on hold during the outbreak [31]. The pandemic has caused significant disruptions in business organizations' operations, especially inventory management. The pandemic brought a great deal of need to inventory management and its lack of critical products, including food and medical equipment, causing a serious deterioration of the global supply chain, which has resulted in revenue loss and unequal demand and supply needs [3]. Organizations are at risk of facing supply-chain interruptions due to deficiencies in raw materials. The production has been decelerated or even suspended, prompting manufacturing firms to wonder how to manage the present inventory situation. PwC conducted opinions for 871 CFOs about the latest developments of COVID-19 and reported that 64% plan to reopen their businesses by using health safety measures, 46% plan to speed up automation or other new ways of working, and 49% believe that their business operations could be expected within a short time if the pandemic ends today, whereas 70% of CFOs are widely worried about the potential adverse business effect of the crisis.

Never before the COVID-19 pandemic has the necessity for accurate inventory management and tracking been so highlighted. There were shortages of necessary products; customers scrambled to collect and manage stock. Moreover, the effects of COVID-19 caused outbreaks in supply chain and inventory management [39]. Furthermore, the supply chain should have a sufficient flow of food and other products in order to prevent inventory shortages. It had become a new reality to meet customer demands that many suppliers had never imagined before, mostly when the huge orders were essential products such as groceries, medicine, and other priority items.

### 4.1.4. Challenges for Workforce Management

The source of COVID-19 and its transfer to humans is unknown; however, the rapid transmission from person to person has been widely confirmed. In light of the dramatic changes caused by the pandemic, changes have been made in organizations, and the changes have to be accepted, and the workforce has to be managed properly [5]. With the limitation on the number of people who could be together at any time, several businesses had taken initiatives to protect their workers and prevent the virus's spread. Industries such as hotels, airlines, tourism, food, and beverages were announcing the massive shutdown [4]. It appeared that most companies had accomplished the challenging job of meeting the basic needs of the safety, security, and stability of their employees in the early stages of this crisis. Organizations had suddenly faced unprecedented challenges in different operational areas such as human resources management. Companies were thinking about how they could work in the long run and short term, while workers and communities were trying to perform while coping with what was happening in their lives. Many organizations faced tough decisions to cut workers, whether through layoffs or other ways.

Management of the workforce plays an important role in helping workers to overcome the challenges posed by unexpected changes both in the workplace and in society [7]. In addition, it is important to shift into the digitalization and collaboration skills of employees to virtual work [40]. The COVID-19 pandemic has already changed the working hours and

the workplace. For example, Twitter and Fujitsu have planned to permanently implement their "work from home" policy, reducing office space use and lessening fixed costs such as rent and electricity bills. Facebook and Google have planned to extend their work from home office policy until 2021, and Amazon has already set a home office policy until October 2021. Thus, employers and employees are suffering from COVID-19 and are seeking guidance on how to proceed.

### 4.1.5. Challenges for Budget Planning

The budget of a business is influenced by several factors, including the environment, strategies for growth, and social relations [41]. The outbreak of COVID-19 has genuinely touched the entire world and has brought with it many uncertainties about the immediate future and what the future looks like as we move forward [42]. In particular, the global economy is facing an inverse demand-supply stock as factories are forced to cut production due to the pandemic, resulting in the separation of the global supply chain network [43]. Business organizations are left wondering where their annual budgets are being impacted and how to plan for this impact as it continues through this year and into the 2021 budget season ahead. At the beginning of 2020, economic activities suddenly stopped or slowed, and as each day passed, the coronavirus pandemic increased the likelihood of a severe recession. Most business leaders have not competed with the recession for the past decade, but COVID-19 has created a new challenge for them; business organizations face financial and practical difficulties while quickly addressing their customers' and suppliers' needs.

Many organizations have to change their budgets and priorities in a mostly different way from what they planned in early 2020 [43]. A survey conducted by Dresner Advisory Services shows that the majority of business leaders (61%) said that the pandemic affected their budgets, and another survey conducted by Gartner on 317 CFOs and finance leaders shows that 62% of the respondents were planning to cut down selling, administrative, and general budgets. By contrast, 38% of the respondents anticipated no reductions now, and 18% intended to slash their expenditure by at least 10% from each category.

### 4.2. Proposed Strategy Using AI and Big Data as Possible Solutions for the Business Operations during COVID-19

#### 4.2.1. Strategy for Smooth Production and Supply-Chain Management

In recent years, managing supply chains has become much more challenging. The COVID-19 pandemic has made this challenge even more difficult. AI-based supply-chain management is expected to be a potent instrument in assisting a firm in overcoming these challenges. An AI-integrated approach that analyzes large amounts of data can solve all business opportunities and limitations from purchase to sale, understand relationships, provide visibility in activities, and help make better decisions [44]. For example, Meituan-Dianping, a food delivery firm, has launched a zero-contact service in major cities in China where the meals are delivered to specific drop-off points, where the customer and delivery workers do not need to interact or meet physically. Earlier, the annual transactions of Meituan were around 450 million. At the beginning of the outbreak, food orders dropped by one-third because customers were afraid to interact with delivery workers. In response, Meituan deployed self-driving vehicles to complete deliveries and a system that allowed customers to pick up their orders from lockers. By February, appropriately 80% of Meituan's deliveries were completed via this zero-contact model.

For another example, Forest Cabin Cosmetics, China saved its business by moving more of it online and harnessing the digital ecosystem of its partners at the early stage of the pandemic. Known mainly for its camellia oil products, the company had to shut down half of its 337 stores and saw revenues drop by 90% at the start of the pandemic. However, beauty consultants at its retail stores quickly became online influencers through their Ding-Talk and WeChat social media. Forest Cabin also streamed sales events through Alibaba's Taobao platform, which was viewed by 60,000 customers. These strategies were so successful that Forest Cabin's revenue rose 120% in March over the same period last year.

Meituan-Dianping and Forest Cabin Cosmetics launched a digital ecosystem for contactless delivery systems with the help of AI-enabled chatbots and big data analytics. AI- and big-data-based platforms helped the two companies to resilience supply-chain management during COVID-19. The two companies' resilient supply-chain management solution included demand-supply forecast modeling, an integrated business plan, and the automation of physical flow—all built on forecast models and interrelationship analyses to better understand the causes and effects in the supply chain. Successfully implementing AI-enabled supply-chain management has enabled Meituan-Dianping to complete 80% of their delivery with a zero-contact model, and Forest Cabin's revenue rose 120% over the same period the previous year. AI and big data present many new opportunities for businesses. The potential use of AI and big data in the logistics system can facilitate firms' smooth business operations [45]. AI has broken down the limited boundaries of human data collection and data processing capabilities for decision making, while big data analytics enable a firm to process more information at faster speeds for timely decision making. Thus, AI and big data analytics could make a firm capable of enriching its supply chain in response to any challenges in the future such as the COVID-19 crisis.

### 4.2.2. Strategy for the Business Model and Innovation

A crisis may bring some new opportunities [36]. The COVID-19 crisis and its associated economic downturn could provide a unique opportunity for some business organizations to redefine their business models and innovation. Business activities have evolved over the decades, from production and supply chains to business intelligence. The rise of AI and big data have fundamentally changed the meaning of innovation, ideas, and inventions by capturing large amounts of data. The adoption and applications of AI technologies and big data can change business models across the world. New business models often arise from turbulent times, making it easier for business organizations to survive after the crisis. For example, Lin Qingxuan, a Chinese cosmetic company, was forced by COVID-19 to close 40% of its outlets, and it lost 90% of its sales. Facing imminent bankruptcy, Sun Liachun, its founder, decided to deploy more than 100 beauty consultants from its outlets to become online influencers. The beauty consultants were charged with using digital technologies to engage customers virtually and execute online sales. With a little support from Ding-Talk's collaboration tools and Alibaba's e-commerce solutions, Lin Qingxuan's revenue in Wuhan increased 200% compared to the previous year's sales revenue.

Another example, Master Kong, an instant noodle producer and distributor in China, faced a sharp drop in supermarket sales by changing its business model because it was challenging to retain its goods stored due to hoarding and the great disruption of transportation by moving much of its sales to JD.com's online channels. Furthermore, in March, Alibaba launched an app named Taobao Deals to support producers without a clear bond with customers, delivering direct-from-the-factory products at extremely affordable rates. Alibaba expects Taobao Deals sales to carry 10 billion new orders to factories throughout China in the next three years.

Artificial intelligence uses data compiled by its computational and mathematical algorithmic models, and the input of human experience makes a decision that a human expert will provide the same decision [46]. Lin Qingxuan, a Chinese cosmetic company, used its trained data with the collaboration of Ding-Talk and Alibaba's AI platforms to make a new business model during COVID-19. Similarly, Master Kong, an instant noodle producer and distributor in China, changed its business model with the collaboration of JD.com and Alibaba's online during the COVID-19 crisis. AI is changing the way of the business models, as evidenced by Airbnb and Uber [46]. AI with trained data can develop a useful business model and support management for effective decision making [47]. Consistent with the previous findings, the two Chinese companies have further proven that the use of AI technologies for digital platforms is the new fuel in any challenging situations such as COVID-19 and competitive marketplaces. Thus, an AI-enable business model could enrich a firm's business operation during any crisis such as COVID-19.

### 4.2.3. Strategy for Inventory Management

Inventory management is the most powerful function of the supply chain process. Access to data has become important for managing inventory. AI and big data analytics have met the human limitation of data collection and data processing. Business intelligence and analytic technologies are used for accuracy because they facilitate easy data collection, analysis, and supply of information designed to help managerial decision making [47]. For example, Ideal, a Chinese Traditional Jeweler, has sold its products for the last two decades. However, when COVID-19 threatened the closure of its business, the company planned to convert to online to survive. Using YouZan, a Saas store management solution on WeChat, Ideal established an AI-based warehouse accessible to its employees and launched an initiative called "Thousand People, Thousand Stores." After proper training, the workers were able to sell their products to their regions online. Each employee runs their store sharing the company's AI-based warehouse. Ideal is now encouraging a more aggressive sales strategy, with commissions from 10% to 50% compared to the previous 3%, and is managing their inventory effectively.

Big data can play a significant role in managing inventory, especially in determining the market's demand side. By shrinking the large volume of data, companies will predict demand better, making the process more proactive from reactive. For example, JD.com, the Chinses e-commerce giant's automated warehouse, increased its sales by 20% with robots and drones during the pandemic [21]. The effort of JD.com represents significant changes in automation, digitalization, and artificial intelligence. The Chinese economy is heading for a significant robotic jump as it removes the touchpoints of human beings that save costs, increase efficiency, and protect public health.

AI-based inventory systems have the ability to accurately interpret supply-demand data, learn from such data, and use those lessons to achieve specific goals and tasks through flexible adaptations [48]. The Chinese Traditional Jeweler company's AI-based inventory system first built a demand prediction model using trained data. The prediction model then estimated what demand would be like for the coming days across the inventory. A jeweler company's AI-based inventory system merged the demand-supply datasets to predict future demand for products. Thus, an AI-based inventory system enables the Chinese firm to make well-informed business decisions, smooth business operations, and increase profit during COVID-19. Similarly, JD.com purchases all products from suppliers for its self-powered AI online store during COVID-19. In addition, JD.com responsibly used AI technology for all subsequent processes, including inventory management, demand-supply scheduling, and after-sales service. JD.com controlled the quality of products and services and increased its sales revenue. Thus, an AI-based inventory management system could enrich a firm business operation during any crisis such as COVID-19.

### 4.2.4. Strategy for Workforce Management

Historically automation happens when economic difficulties are combined with maturing technology [21], and COVID-19 is the best example of "an automation-facing event." The use of automation systems in the Chinese economy has made China a leader globally though it has a large-scale workforce. Automation affects some jobs, but it still creates several new jobs. For instance, new digital data infrastructure centers, different software, 5G equipment, and robotic operations and repairs will require human workers. Similarly, human-centric information collection and labeling require humans an "energy" that empowers AI and supports an automated economy. Robotic process automation (RPA) can reduce disruption by allowing companies to remain linked across teams and networks, retain customer satisfaction, and provide stability during uncertain times. For example, Mater Hospital in Ireland states that the COVID-19 test results must be entered into different databases and regularly reported to the IPC department—a process that demands three hours of input every day, and the systems were used during the SARS outbreak. Now, the hospital has adopted RPA robots and UiPath to optimize COVID-19 test results. UiPath robots sign in the hospital's system, apply the disease code, and enter the test result. The

robots save three hours of work every day in the hospital, which are important hours that can be spent interacting with patients instead of data entry.

Another example, the Hongshan Sports Center in Wuhan, China, opened a field hospital with robots where the pandemic started. The facility is a project of China Mobile, Wuhan Wuchang Hospital, and Cloud Minds. Patients entering were tested for fever by attached 5G thermometers. Patients had smart bracelets and rings integrated with the AI app of Cloud Minds to track their diseases' critical signs, including temperature, pulse rate, and blood oxygen. Doctors and nurses have used these instruments to spot early symptoms of infection. Meanwhile, other robots supplied patients with food, beverages, medicine, information, and amusement by dancing, and other autonomous droids sprayed disinfectant and washed the floors. The field hospital is one of many in Wuhan that treated over 20,000 patients if regular hospitals became overburdened.

Robots can have an equal impact by enhancing the need for employees to maintain skills and gain new and additional complex skills [49]. Going back to our examples from above, to provide better service to patients and enhance employees' skills, the two hospitals have adopted robots during COVID-19. Robots and automation processes via AI are able to successfully support the repetitive work of human workers [50]. The Hongshan Sports Center in Wuhan used robots to determine the temperature, pulse rate, and blood oxygen of patients. Similarly, robots supplied patients with food, beverages, and medicine. Robots boost efficiency in the workplace. In Mater Hospital in Ireland, robots save three hours of work every day, and the Hongshan Sports Center in Wuhan, with the help of robots, treated over 20,000 patients during COVID-19. Thus, business leaders gathering experiences from the above two examples can integrate robots to enrich business operations during any crisis such as COVID-19.

4.2.5. Strategy for Budget Planning

In regular times, budget-planning teams typically use driver-based models to analyze, forecast, and identify the root causes of budgeting. In this unprecedented pandemic time, the budget-planning team needs a new systematic approach that will alert them on the options available to come out of the fast-growing crisis. By embracing big data and AI, companies can identify vast opportunities within the crisis. Applying data science techniques and big data analysis can improve the organization's budgeting process [51]. Data science and machine learning algorithms could predict the budget, which could help compare future outcomes and modify the budgeting model. Budgeting is no longer a simple spreadsheet exercise in which the sales and marketing manager converts a collection of sales figures into a financial table. Instead, it adds more stability to the operational projection based on multiple internal and external information sources. Data is now driving business. For example, companies can forecast probabilistic plans with sales data and develop remedial measures to counter its negative impacts.

AI, machine learning, and predictive analysis can pave the way for comprehensive business-centered data analysis to maximize sales. For example, Toyota Motor Corp. saw a sharp fall in sales revenue in China due to the coronavirus outbreak. Toyota's new vehicle sales in China, including Hong Kong and Macau, dropped 70.2% from a year earlier to 23,800 units in February, according to Toyota Motor (China) Investment Co. The company's sales also fell 15.9% in March when the pneumonia-causing virus spread stifled the Chinese economy. Fortunately, the company's sales have started growing since May. With the analytic help of big data, the car manufacturer said that its company is on a recovery path in China despite being severely hit by the virus. The national sales for Toyota, the largest automaker, have declined by 20.1% compared to the previous year's 166,300 units, after a 0.2% increase in April.

Big data analytics help companies analyze customer needs and preferences. These factors enable a firm to smooth operation and to make more revenue. Going back to our example of Toyota Motor Corp in China, it increased profit by 0.2% using big data analytics during COVID-19. Big data has been a crucial factor in improving business

organizations' competitiveness in a dynamic market environment. With more data generated and processed than ever, reliability, accuracy, and detailed processing have enhanced organizational efficiency. Thus, big data have opened up a new area where businesses can improve their strategic decision-making process to enrich smooth business operation during any crisis such as COVID-19.

## 5. Discussion

### 5.1. Key Findings

The COVID-19 pandemic has dramatically changed the world, not only in the healthcare system but also in many aspects of human life, including transport, tourism, education, the business operating system, etc. Over the last few months, business leaders have faced various short-term and long-term challenges in business operations. However, some firms have proved that AI and big data can support organizations for their smooth operation during the pandemic. This study aims to discuss major challenges that business firms face to operate a business effectively during COVID-19 and its expected solutions using AI technologies and big data. Specifically, this study first identifies five important challenges for effective business operation during COVID-19: production and supply chain disruption, appropriate business model selection, inventory management, budget planning, and workforce management. The paper outlines possible solutions for five challenges by explaining eight business cases of firms that used AI and big data during this crisis.

AI enables the quick analysis of data in order to provide an appropriate answer to any business issues [52,53] and make accurate forecasts of customers' needs. Thus, our sample companies using AI and big data increased their revenue during the outbreak. For example, JD.com, the Chinses e-commerce giant's automated warehouse, increased its sales by 20% with robots, while the Toyota corporation used big data analytics for preparing the budget during this outbreak. Similarly, two cosmetics companies, Lin Qingxuan and Forest Cabin, increased their revenue by 200% and 120%, respectively, by employing AI technologies. On the other hand, Meituan-Dianping, a food delivery company, completed 80% of deliveries through a human contact-less model, and Master Kong, a noodles producer and distributor, fulfilled 10 million new orders using AI and big data technologies. Thus, our findings are consistent with AI and Big data applications in every aspect of businesses [54], the supply chain [55], and manufacturing [56].

The overall findings reveal that the crisis of COVID-19 could inspire a movement for more dynamic leadership using AI and big data in business operations. To tackle the challenges and risks thrown by COVID-19, firms should develop some strategies. First, a short-term business model should be prepared at the first stage. Second, a long-term comprehensive improvement in the business strategy provides resilience and permits proactive alterations to improve performance. While the consequences of COVID-19 may undoubtedly be widespread, the study emphasizes the latest AI-related technologies and big data analytics on selected topics relating to decision making for guiding firms' operation in the current business environment.

### 5.2. Theoretical Implications

The study contributes by extending the RBVT in the existing literature by identifying AI and Big data as invaluable resources and effective strategic tools for solving tentative managerial problems. This study argues that using AI and Big data as invaluable resources creates new opportunities to utilize a firm's full capacity in a competitive advantage and risk mitigation weapon in any challenging environment such as COVID-19. Researchers also explore several cases that state how AI and big data can help take timely actions in any crisis. It also adds to the literature by providing insights into the interplay between managerial decision making and crisis management in AI and big data contexts. However, lessons from the COVID-19 pandemic show that many firms are now doing better using the AI technologies and big data that can play a significant role in their organizational management, particularly the resilience of supply chain management, effective and effi-

cient budgetary control systems, effective inventory management, a good workforce, and innovative business model development.

Our findings contribute to the prior literature by identifying the positive effect of AI and big data usage in supply chain management, inventory management, business models, workforce management, and budgeting during any crisis such as COVID-19. Consistent with the findings of [57,58], we extend the literature on how AI and big data usage in business can enrich a firm's smooth operations. Our findings imply that AI and Big data can be thought of as a coping strategy of business that becomes effective for resilient business operations during any crisis such as COVID-19. To the best of our knowledge, these relationships have never been investigated in prior literature on how a firm enriches its business operations using AI and big data.

Our study contributes to the existing literature in general by identifying additional predictors. Prior literature found that digitalization has a significant effect on enhancing employees' task performance and efficiency as well as firms' performance [59]. In contrast, technology dissemination, adoption, and acceptance all involve aspects of human dynamics by which new artifacts sometimes become a failure in the social and business processes of organizations [60]. Therefore, our study contributes to this body of literature by showing a direct relationship between the usage of AI and big data by a firm and smooth business operations (e.g., supply chain management, business models, inventory management, workforce management, and budgeting) during any crisis such as COVID-19. Our findings imply that AI and big data usage in business activities enhance effective business operations in any crisis in the future.

*5.3. Practical Implications*

The findings of our study indicate that AI and big data usage in business is likely to enrich a firm operation during COVID-19. Technologies have a pivotal role in business, different activities in effective operations, and business value [61]. When a firm engages in AI and big data usage to enhance its intrinsic value and to help business activities, it is a signal for the workforce that employees will also adapt well, and thus, the firm will operate effectively and enhance its performance [62]. As AI and big data usage in business operations boost employer attractiveness for effective operations, firms that aim to increase their performance by engaging in AI and big data usage need to certify that their AI and big data usage policy is designed within the matching line of employees' desire to firms for helping better operations and performance. Therefore, our research will help business leaders adopt an AI and big data usage policy. Its implementation should be carefully planned for better business operations and performance to handle any crisis such as COVID-19.

Over the years, the world has moved towards an automated future, and AI and big data usage in business activities is seen to be the way of the future [9]. The COVID-19 pandemic has further enhanced the importance of AI and big data usage in business activities. For example, AI-driven supply chain management allows businesses to forecast demand spikes and declines accurately and adjust material volumes and routes. Additionally, AI and big data can be used to gather more comprehensive data that may assist the sales team in forecasting more precise delivery timings and inventory adjustments. As a result, businesses are able to provide superior customer service to present and prospective consumers during times of crisis, such as COVID-19. Similarly, AI has changed businesses, as proven by Uber, Flip cart, e-Bay, and Amazon, which have incorporated its use to instrument state-of-the-art AI-enabled business models [46]. Thus, AI-driven work events have been changing the way businesses perform. Our research model will help business leaders to adopt AI and big data in their firms for effective business operations to any crisis in the future. The superior accuracy of AI in decision making has revolutionized and reshaped the way business is performed [63]. For example, AI algorithm-based budgeting has overcome the boundaries of limited human data processing capacity and has made more reasonable decisions in a timely [64]. Thus, our study will help the business policymaker make a

time-driven decision of using AI in inventory management and budgeting for effective operations during the COVID-19 crisis

Finally, AI and big data can develop and evolve efficient management control, decision-making, and budgeting processes [50]. Therefore, business leaders, using AI and big data, can quickly adjust their business strategies to overcome any challenges such as COVID-19, plan the underlying assumptions, take a playful view, and apply a focused-on-digitalization trend. Overall, this study may have policy implications for managers in recognizing the vital area of business operation, where big data and AI can significantly improve performance during any similar crisis in the future.

## 6. Conclusions, Limitations, and Future Research

How can a firm enrich its business operations using AI and big data during COVID-19? To shed empirical light on this question, the study identifies five critical challenges of business during COVID-19: production and supply chain disruption, appropriate business model selection, inventory management, budget planning, and workforce management. The paper outlines possible solutions for five challenges by explaining eight business cases as examples from business firms that used AI and big data during this crisis. These business cases show that AI and big data are crucial driving forces for effective business operation in this challenging environment. The outcomes of this study provide valuable insights into contemporary social science research and business management with AI and big data application as a business response to any crisis in the future.

This paper is limited to discussing only the theoretical aspects of business processes, focusing on AI and big data in crisis management. Future research may explore the manager's opinions regarding AI and big data in a pandemic situation. Particular business cases are taken as examples to link the challenges and possible solutions to business processes during the COVID-19 pandemic. Potential researchers can gather qualitative and quantitative data via interviews, focus group discussions, panel discussions, surveys, and databases. An industry-wide study may be conducted to examine the efficiency of AI-related technology and big data in a particular business firm. The quantitative/empirical study can be conducted with a broad set of expert opinion responses for the generality of research outcomes. Different economies-based studies can be pursued to test the role of AI and big data in business processes during the pandemic, such as in developed, developing, and underdeveloped nations. Lastly, potential researchers can make several comparative studies between pre-COVID-19 and post-COVID-19 situations and apply AI-related technology and big data to business processes.

**Author Contributions:** Conceptualization and methodology: M.I.B.; validation, review, editing, and supervision: Y.C. All authors have read and agreed to the published version of the manuscript.

**Funding:** National Natural Science Foundation of China under (Grant #72172132) and the Xiamen University COVID-19 Emergency Response Research Funds.

**Institutional Review Board Statement:** Not applicable.

**Informed Consent Statement:** Not applicable.

**Data Availability Statement:** Not applicable.

**Conflicts of Interest:** The authors declare no conflict of interest.

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
