# Peer review of "Turning Crisis into Opportunities: How a Firm Can Enrich Its Business Operations Using Artificial Intelligence and Big Data during COVID-19"

_sustainability, doi:10.3390/su132212656_

Round 1

Reviewer 1 Report

After reading this manuscript, there are some problems that need to be explained. I think it needs a major revision before it can be accepted.

  1. As the Abstract mentioned, the five major business challenges were identified by researchers, but it looks like this contribution is made in this research in the subsequent chapters. I think it need to point out clearly what this research’s original contributions are. Moreover, the paper lacks methodology explanation for identifying these challenges if it’s the main contribution of this manuscript, and the original contribution is inadequate if this contribution is not part of the study.
  2. The analysis of cases and strategic solutions derived are shallow, maybe in-depth analysis and research will enhance the logic chain and improve the research meaning.

Reviewer 2 Report

This manuscript explains the challenges faced by different industries. Even though the authors claim that they identified the challenges and how they are addressed using Artificial Intelligence and Big Data techniques, I find the paper lacks rigor and has no technical depth. Most of the statements explained using the case studies do not have citations, proofs, or analyses explained in this manuscript. Here are my major concerns:

  1. From the research framework (Figure 2), I understand the only research that went through is via search engines. In such cases, detailed data, including the links considered, should be provided to readers. However, this manuscript fails to give any details are data. 
  2. The authors chose eight companies to explain how AI and Big data helped to improve the business model. But, nowhere in the text did the authors explain AI/Big data techniques or methodology used. There were unclear or vague statements. For example, Lines 452-454: “The potential use of the technology in the production and logistics system can facilitate firms’ manufacturing and capturing data processes” – no explanation on what aspects are considered, what are the new inputs provided to the technology due to pandemic? 
  3. There were statements such as “AI can supply information to develop a useful business model”. See Line 492. AI cannot supply anything. AI works based on the input the users provide and train the model to learn the pattern.  
  4. Case studies are too general: how did AI play a role when the companies moved to the online platform? How did the authors get the profit numbers? If it is available on google, where are the citations?
  5. Section 4.2 title says the proposed strategy. However, I see only general statements, and there is no takeaway for future business, thus failing to justify the need for this manuscript. 
  6. Figure 1, Figure 2, and Table 1 are just a set of general statements without any proof. When providing flowcharts, please give a sequential explanation. Especially for figure 2, nowhere the authors mention what research objective-1 and research objective-2 are.

Reviewer 3 Report

I suppose the issue of new technologies is important and influencing business seriously during the time of pandemia. However the study does not bring any breaking insights into this issue, indeed. The weakest element of this study is the methodology employed. Thanks for this methodology research achievements look very essayically.

I suggest authors to consider some tips when working on study improvement: 1. give some justification why these two technologies, why not other from so called I4.0 are in their scope of interests, 2. when presenting research output you need to be more specific, more concrete info, more facts from studied cases, more focus on considered two technologies (you mention now many more technologies), 3. your discussion and conclusions are poor, rework these sections noticeably pls  

Round 2

Reviewer 1 Report

Paper can be accepted for publication.

Reviewer 2 Report

The authors addressed all the comments suggested in the previous revision. 

Reviewer 3 Report

Dear authors, I see you provided a substantial improvement. Thank you.